**Article** https://doi.org/10.1038/s41467-023-37393-0

# Transmembrane signaling by a synthetic receptor in artificial cells

Ane Bretschneider Søgaard [1,2,3], Andreas Bøtker Pedersen [1,3], Kaja Borup Løvschall[1], Pere Monge[1], Josefine Hammer Jakobsen[1], Leila Džabbarova [1], Line Friis Nielsen[1], Sandra Stevanovic[1], Raoul Walther [1] & Alexander N. Zelikin [1,2] ✉

Signal transduction across biological membranes is among the most important evolutionary achievements. Herein, for the design of artificial cells, we engineer fully synthetic receptors with the capacity of transmembrane signaling, using tools of chemistry. Our receptors exhibit similarity with their natural counterparts in having an exofacial ligand for signal capture, being membrane anchored, and featuring a releasable messenger molecule that performs enzyme activation as a downstream signaling event. The main difference from natural receptors is the mechanism of signal transduction, which is achieved using a self-immolative linker. The receptor scaffold is modular and can readily be re-designed to respond to diverse activation signals including biological or chemical stimuli. We demonstrate an artificial signaling cascade that achieves transmembrane enzyme activation, a hallmark of natural signaling receptors. Results of this work are relevant for engineering responsive artificial cells and interfacing them and/or biological counterparts in co-cultures.

Mechanisms of transfer of information across the lipid bilayer of a cell membrane are among the most important evolutionary adaptations. These mechanisms allow the cells to sense the external environment and to communicate within multicellular ensembles. Transfer of information across the lipid bilayer relies mainly on transmembrane proteins, called signaling receptors, which perceive information at the cell surface and communicate it to the cell interior, leading to cellular responses[1–3]. With the exception of nuclear receptors, transfer of information is performed through sealed membranes whereby the activator molecule binds to the receptor but does not enter the cell. In recent decades, design of artificial receptors has become highly important for biomedicine. One example is the chimeric antigen receptor technology, which is rapidly transforming the landscape of possibilities for cancer intervention and treatment[4]. Cell engineering is particularly successful when artificial receptors are orthogonal to their natural counterpart, a feature that ensures specificity of receptor activation[5]. Currently, in these applications, receptor design is based

on proteins, that is, reusing the tools of nature[5–7]. In stark contrast, design of artificial, synthetic signaling receptors, and transfer of information through sealed biomolecular membranes using tools of chemistry remains to be a grand fundamental challenge with only few successes[8].

The main area of application for artificial receptors is in the design of synthetic cells (syncells)[9]. This field of science is highly interdisciplinary and receives contributions from the diverse sub-disciplines of chemistry, physics, and molecular and cell biology, as well as neighboring areas of research[10,11]. Syncells are academically intriguing and also hold promise for applications in synthetic biology, biotechnology, and biosynthesis[10,12]. Towards this end, cell-like vesicles have been successfully engineered using lipids and/or polymer molecules[13]. Encapsulated catalysis has been highly successful with documented inspirational examples of enzymatic reactions, transcription, and translation, including the use of syncells for protein synthesis and delivery in vivo[14,15]. Co-culture of artificial and natural

[1]Department of Chemistry, Aarhus University, Aarhus C, Denmark. [2]iNano Interdisciplinary Nanoscience Center, Aarhus University, Aarhus C, Denmark. [3]These authors contributed equally: Ane Bretschneider Søgaard, Andreas Bøtker Pedersen. ✉e-mail: zelikin@chem.au.dk

cells has proven to be a major step towards successful tissue engineering[16,17]. A particularly challenging aspect in the development of multicellular assemblies has been the communication between syncells within their ensemble or with natural counterparts[18]. It has been successfully accomplished using several inspirational ways, yet in most cases it relied on *diffusive* communication whereby the inner volumes of syncells exchange solutes as large as nucleic acids and proteins[19–21]. Engineering the nature-mimicking responsive behavior in artificial cells using synthetic signaling through *sealed* (biological) membranes would be a significant advancement for biomimicry, yet successes on this avenue are few.

One class of artificial signaling receptors, initially designed by Hunter and Williams et al.[22,23] and later adopted by Schrader et al.[24], relies on the toolbox of chemically induced dimerization[5] and uses membrane-spanning cholesterol dimers. A dimerization event at the exofacial surface (due to e.g., oxidation of thiols into a disulfide[22] or Cu$^{2+}$-mediated bridging of two monovalent ligands[23]) evoked proximity of the two transmembrane molecules and ensued dimerization of the endofacial termini of these molecules, without compromising integrity of the lipid bilayer. Typical results in these studies included a release of a UV-active molecule[22] or an energy transfer event between the dimerizing "sub-units"[24]. Another class of artificial receptors was designed by Hunter et al.[25–27] using molecules that exhibit controlled "bobber"-like translocation across the lipid bilayer. In these cases, receptor activation (by a change in solution pH[25], the presence of copper ions[26], or a competitive ligand displacement event[27]) ensued a cross-membrane movement of the receptor molecule, which resulted in an exposure of a Zn-coordinating ligand as a metalloenzyme mimic. Finally, Clayden et al.[28] developed peptide foldamers that exhibit in-membrane conformational change, mimicking the performance of natural receptors, in response to an enkephalin agonist. Each of these results is highly important, but the field is still in its infancy.

Self-immolative linkers (SILs) enable stimuli-responsive traceless drug release from a prodrug in response to an activator, which can be physical, chemical, or biochemical (enzymatic), and are highly successful in both academia and in industry[29–34]. In our recent work, we designed cell surface-anchored prodrugs as artificial apoptosis-inducing receptors: exofacial activation of the prodrug ensued decomposition of the SIL and release of the secondary messenger molecule, which was a potent toxin[35]. These results led us to recognize that SILs can serve as a chemical mechanism for signal transduction across sealed biological membranes, thus addressing the challenge of receptor mimicry in syncells.

Here, we present the design of SIL-based synthetic receptors mediating artificial transmembrane signaling. These receptors are designed to exhibit similarity to the natural counterparts in the most essential characteristics, namely in having an exofacial trigger to capture the input signals, in being membrane-anchored, and in featuring a releasable, secondary messenger molecule (Fig. 1a). The secondary messenger is chosen such that it can activate an encapsulated enzyme, thus fulfilling the mimicry of downstream signaling, which is the hallmark of receptor performance in nature. We show that the adaptable nature of SILs as a signal transduction mechanism enables chemical diversity of synthetic receptors, specifically with regards to the receptors activator molecules. To realize this potential, we (i) designed a modular platform for the synthesis of various artificial chemical receptors that feature 1,6-benzyl elimination as the mechanism of signal transduction; (ii) accomplished the synthesis of synthetic receptors that can be activated by chemical and biochemical stimuli; (iii) validated transmembrane signaling using synthetic receptors in syncells based on liposomes; and (iv) confirmed the transmembrane (non-diffusive) mechanism of signaling. We believe that the highest achievement of our work lies in having established artificial transmembrane signaling that connects a biologically relevant input to a biologically relevant output, whereby the released secondary messenger activates an encapsulated enzyme, as is the hallmark of signaling in nature.

## Results and discussion
### Receptor design
The synthetic receptor molecules proposed herein are engineered using *p*-hydroxybenzyl alcohol as the core structural element that comprises the SIL, that is, the signal transduction mechanism (Fig. 1b). The receptor molecule is an amphiphile: it contains a polar fragment that favors aqueous, exofacial localization, to engage with the receptor-activating stimulus, and it contains a hydrophobic part for anchoring within the lipid bilayer. The exofacial part comprises the trigger group, removal of which initiates the decomposition of the receptor molecule, ensuing release of the secondary messenger molecule. This trigger group is therefore installed at the phenolic end

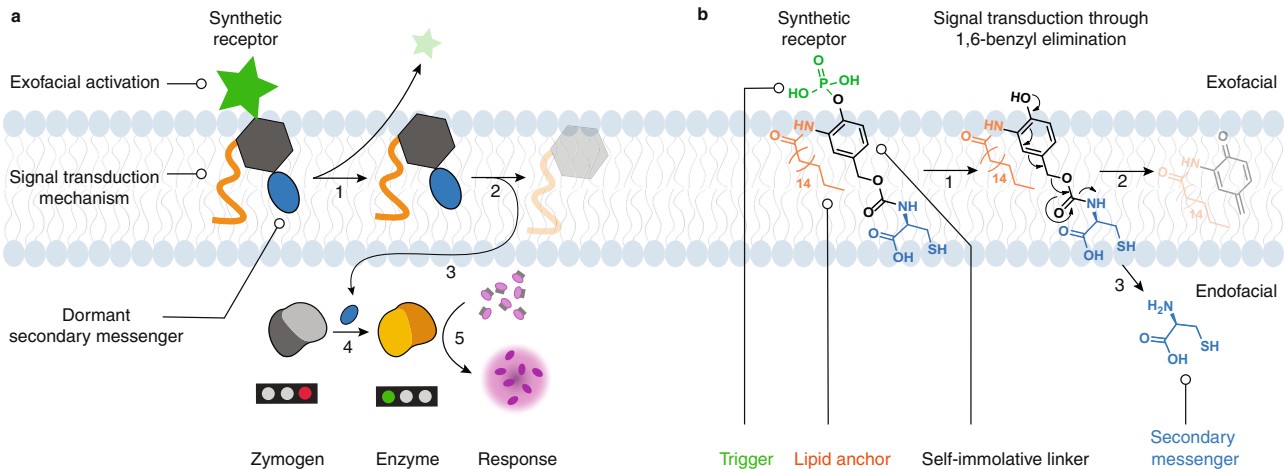

**Fig. 1 | Schematic and chemical illustration of the proposed concept of transmembrane signaling using our synthetic receptors. a** Schematic showing (1) receptor activation at the exofacial side of the membrane, (2) self-immolation and ensuing generation of the secondary messenger, which (3) enters the inner compartment (4) and activates the chemical zymogen; (5) leading to a secondary signaling event and signal amplification via enzymatic catalysis. **b** Specific chemistry

realized in the synthetic receptor molecule: it features an exofacial hydrophilic phosphate ester moiety for receptor activation using the corresponding activating enzyme (alkaline phosphatase, ALP); lipid-modified p-hydroxybenzyl alcohol as a self-immolative linker which acts as the chemical mechanism of signal transduction via 1,6-benzyl elimination; natural amino acid L-Cys is released and acts as a secondary messenger.

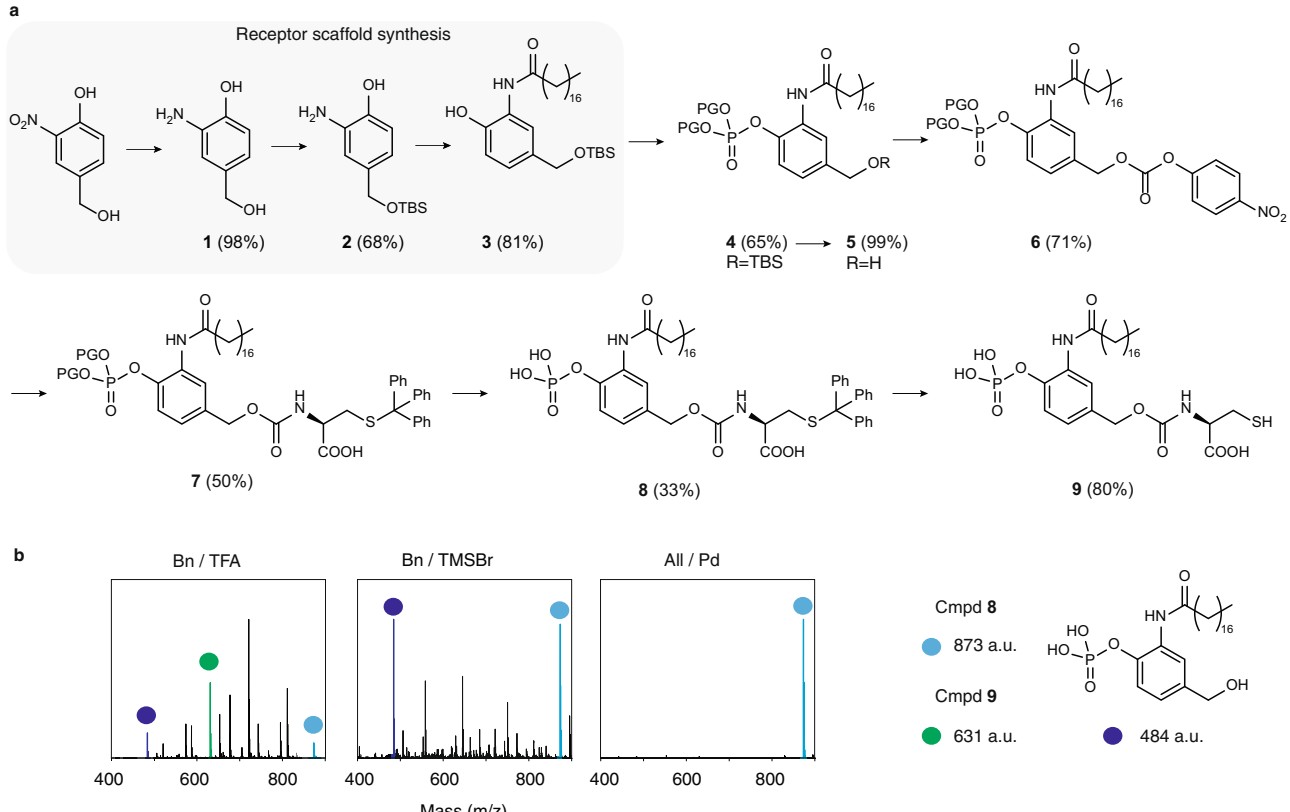

**Fig. 2 | Synthetic pathway to phos-EAR molecule. a** Schematic illustration of synthesis of the receptor molecules, starting from 3-nitro-4-hydroxybenzyl alcohol, to the synthesis of a modular receptor scaffold **3**, to the phosphatase-specific enzyme-activating receptor **9** (Phos-EAR). PG = protecting group **b** HRMS data illustrating deprotection of the benzyl-protected compound **7** using TFA or tri-methylsilyl bromide (TMSBr), and an allyl-protected **7** through Pd-catalyzed deal-lylation. For detailed reaction conditions, see Supplemental Methods.

of the SIL. For lipid bilayer anchoring, the *ortho*-position to the phe-nolic alcohol in the SIL is used to install a suitable functionality, in our case a $C_{18}$-aliphatic chain. Finally, the benzylic end of the SIL is used to install a releasable secondary messenger molecule. Transmembrane signaling events in nature culminate in enzyme (de)activation, which comprises the downstream signaling cascades[2,3]. To mimic this, we used a cysteine protease, papain. The activity of this enzyme is blocked by a chemical modification of the catalytic thiol group in the active site, namely by conversion into a disulfide[36,37]. The catalytic activity of the chemical zymogen is restored via thiol-disulfide exchange with another thiol-containing molecule—thus presenting an opportunity to achieve chemically triggered activation of enzymatic catalysis[37]. This notion suggested that the secondary messenger released by the receptor upon its exofacial activation must be a thiol-containing solute. To complete biological relevance of the artificial receptor designed herein, we chose to use an amino acid L-Cysteine (L-Cys) as a natural thiol-containing molecule (Fig. 1b).

The first receptor realized synthetically in our work contained the highly polar phosphate ester as an exofacial SIL triggering group. The synthetic path to the receptor molecule consisted of a total of 9 steps (Fig. 2a, see Supplemental Methods for details). First, we synthesized a modular receptor scaffold, namely *p*-hydroxybenzyl alcohol functio-nalised with a lipid anchor, and *tert*-butyldimethylsilyl (TBS) protected at the benzylic end (**3**). The phenolic end of the scaffold was then converted to a protected phosphoester (**4**), before the benzylic end was converted in two steps into an activated carbonate (**6**). The latter was reacted with *S*-Trityl-L-Cysteine (*S*-Trt-Cys) to afford the receptor molecule-bearing protecting groups (**7**). Removal of the protecting group of the phosphoester proved to be the major hurdle and required

several rounds of optimization (judicious tuning of reaction conditions and choice of protecting group) to accomplish. Initial attempts to access the desired receptor molecule employing ethyl- or benzyl-functionalised phosphoesters were met with failure. Specifically, attempts to remove the benzyl ester using trifluoroacetic acid (TFA) resulted in a crude mixture of compounds that contained diverse variants of deprotection of the phosphoester, cleavage of the carba-mate linkage, and/or removal of the *S*-Trityl (S-Trt) protecting group (Fig. 2b). Deprotection was also conducted using trimethylsilyl bro-mide and this approach too resulted in a complex mixture of com-pounds. Successful synthesis was accomplished using a different protecting group strategy, namely diallyl-phosphoester. In this case, the phosphoester compound **7** was deprotected through Pd-catalyzed deallylation which afforded the *S*-Trt protected receptor molecule **8** without associated products of degradation. Finally, removal of the *S*-Trt protecting group afforded the desired receptor molecule **9**, termed Phos-EAR, for phosphatase-specific <u>E</u>nzyme-<u>A</u>ctivating <u>R</u>eceptor.

The design of receptor molecules around the modular scaffold **3** has an advantage of offering chemical diversity, specifically through the choice of the triggering group at the phenolic end of the SIL (Fig. 3a). To demonstrate this, we synthesized two more receptor molecules, containing glucose or glucuronic acid as triggers for receptor activation. For the β-glucosidase (GLU) specific receptor, per-acetylated glucose was converted into the corresponding glucosyl bromide and thereafter coupled to the scaffold molecule **3** to afford compound **10**. Subsequent steps (removal of the benzylic silyl ether, synthesis of the nitrophenyl carbonate, and carbonate exchange to *S*-Trt-Cys-bearing carbamate) were conducted via the protocols much

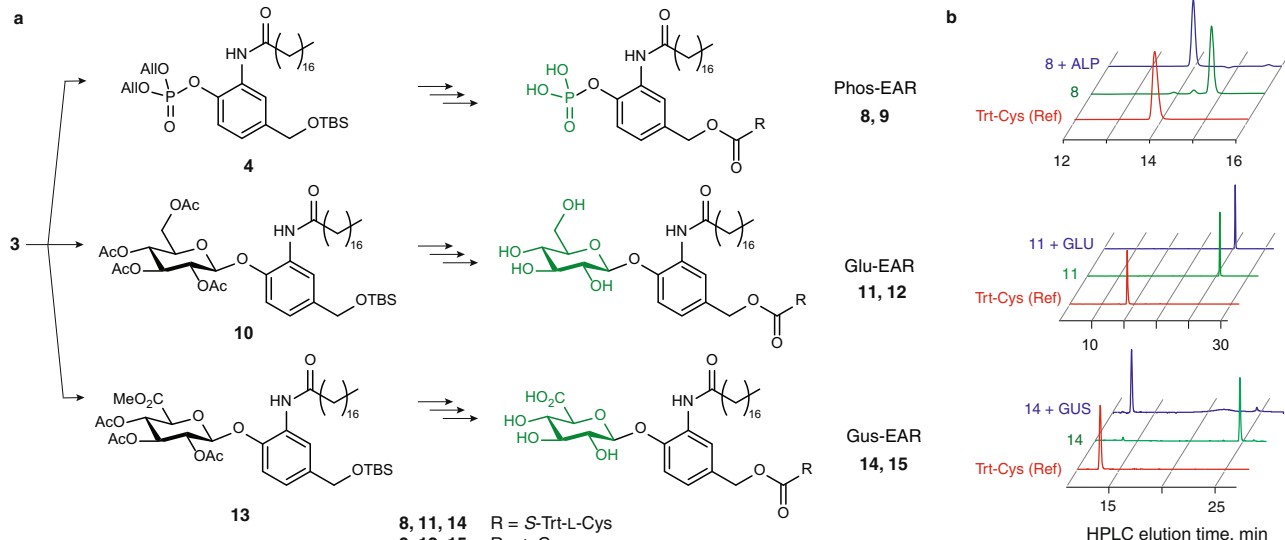

**Fig. 3 | Diversity of the synthetic receptor molecules. a** Schematic illustration of the synthesis of three different receptor molecules for ALP, GLU, and GUS-mediated activation: Phos-EAR (**9**), Glu-EAR (**12**), and Gus-EAR (**15**), respectively. **b** HPLC data illustrating the release of the secondary messenger S-Trt-Cys in response to the nominated enzymatic trigger. ALP = alkaline phosphatase, GLU = β-glucosidase, GUS = β-glucuronidase. Raw data are provided as the Source Data file.

similar to those employed for the synthesis of the Phos-EAR (**9**). Lastly, deacetylation was conducted to obtain the S-Trt-protected receptor molecule **11**; subsequent S-Trt deprotection of **11** afforded the GLU-specific receptor molecule (**12**, Glu-EAR). The β-glucuronidase (GUS) specific receptor molecule (**15**, Gus-EAR) was synthesized in analogous fashion using glucuronic acid instead of glucose. Uronic acids are notoriously poor glycosylation reagents, which explains the low yield of the glycosylation step (4%). Nevertheless, both the scaffold molecule **3** and the glycosyl bromide are readily available (commercially and/or synthetically), to obtain sufficient quantities of compound **13**. Deprotection of the per-acetylated methyl glucuronate was conducted via a two-step one-pot procedure to obtain S-Trt protected GUS-specific receptor molecule **14**, and subsequent removal of the S-Trt protecting group afforded the desired Gus-EAR molecule **15**. We note that synthetic diversity realized herein only concerns the receptors exofacial triggering groups (phosphate, glucose, glucuronic acid). Nevertheless, prior efforts from us[31,38,39] and others[40] have documented numerous examples of triggered 1,6-benzyl eliminations that results in the release of drugs, imaging reagents, or other reporter molecules, each of which becomes the secondary messenger to suit the wanted application. In our work, we focused on L-Cys, specifically to achieve activation of an encapsulated enzyme from its disulfide-based chemical zymogen using the natural amino acid as a secondary messenger.

## Enzyme-mediated receptor activation in solution

To validate the enzyme-mediated receptor activation and the subsequent release of the secondary messenger in solution, we used the S-Trt-protected compounds **8**, **11**, and **14**, to capitalize on the strong UV–vis absorbance of the Trt-group. Surprisingly, the treatment of the Glu-EAR precursor **11** with GLU did not afford the expected product S-Trt-Cys and as such we observed no enzymatic reaction taking place (Fig. 3b). To resolve this finding, we synthesized two more molecules, namely glucosylated p-hydroxybenzyl alcohol with and without the $C_{18}$ aliphatic group in the meta-position. HPLC monitoring of the catalysis revealed that GLU performed the expected glycolysis on the O-aryl glucoside of p-hydroxybenzyl alcohol, but not on its counterpart with the $C_{18}$ aliphatic group (Supplementary Fig. 1). These data indicate an unexpected finding, that GLU-mediated cleavage is prevented by the $C_{18}$-aliphatic anchor in solution. For the

Phos-EAR precursor **8**, in full agreement with expectations, HPLC monitoring revealed that the addition of alkaline phosphatase (ALP) afforded release of S-Trt-Cys with full conversion (Fig. 3b). The same level of success was registered for the Gus-EAR precursor **14** where the addition of the GUS enzyme achieved the exhaustive release of S-Trt-Cys (Fig. 3b). Taken together, the results presented in Figs. 2 and 3 illustrate the synthesis of a modular scaffold for the design of artificial receptor molecules with 1,6-benzyl elimination as a mechanism of signal transduction.

## Receptor performance in artificial cells

Giant unilamellar vesicles (GUVs) were used to visualize receptor anchoring into the lipid bilayer (Fig. 4a). Toward this end, GUVs were incubated with Gus-EAR and fluorescein maleimide. The thiol-reactive dye exhibited negligible association with the lipid bilayer in the absence of the receptor molecule, and no increase in fluorescence was observed compared to the GUVs without added dye. In contrast, upon addition of the thiol-reactive dye to the GUVs that contained Gus-EAR, the lipid bilayer exhibited a pronounced level of fluorescence, indicating covalent reaction of fluorescein maleimide with the L-Cys thiol within the structure of the receptor molecule.

Validation of receptor performance was performed in liposomes based on egg-yolk L-α-phosphatidylcholine. Lipid hydration was performed in solutions containing the disulfide-based chemical zymogen of papain and the fluorogenic protease substrate, $N_{α}$-Benzoyl-L-arginine-7-amido-4-methylcoumarin (Arg-AMC). The receptor molecule Phos-EAR **9** was administered onto liposomes using a small volume of a concentrated DMSO stock, such that the final DMSO content did not exceed 1 vol.%. The final bulk concentration of Phos-EAR was 2 µM. Under these conditions, we observed no increase in fluorescence in solution (Fig. 4b), indicating that addition of EAR does not lead to spontaneous activation of the encapsulated papain zymogen. The EAR molecule contains a thiol functionality and receptor dormancy is conditioned by the placement of the zymogen and the EAR in two different phases, liposome lumen vs the lipid bilayer. This separation is very efficient and in the presence of dormant EAR, the encapsulated zymogen registered minor if any "resting" signal (Fig. 4b). External addition of the ALP enzyme to the liposomal preparation afforded a pronounced increase in fluorescence. This is indicative of ALP activity on the bilayer-anchored Phos-EAR to remove

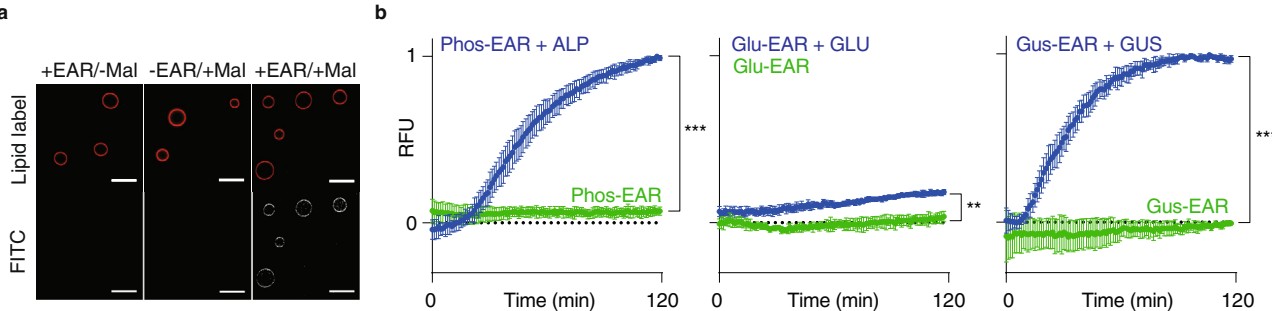

**Fig. 4 | Lipid bilayer anchoring and proof of concept signaling. a** Confocal fluorescence microscopy imaging illustrating association of the GUS-specific receptor molecule with the lipid bilayer of the GUVs; EAR = enzyme-activating receptor, Mal = fluorescein maleimide, scale bars are 20 μm. **b** Kinetic data illustrating evolution of fluorescence in suspensions of liposomes that contain a disulfide-based zymogen of papain and its specific fluorogenic substrate (Arg-AMC); liposomes were equipped with 2 μM Phos-EAR, 20 μM Glu-EAR, or 20 μM Gus-EAR and then exposed to the corresponding activating enzyme, alkaline phosphatase (ALP), β-glucosidase (GLU), or β-glucuronidase (GUS). **b** The data are expressed relative to the max value of the EAR + enzyme sample in Phos-EAR and Gus-EAR, respectively, the Glu-EAR samples are relative to the Gus-EAR + GUS. **b** The data are based on three replicates with triplicates ($N = 3$, $n = 3$) presented as mean ± SD. RFU = relative fluorescence expressed in arbitrary units; statistical significance at endpoint was calculated via an unpaired two-tailed $t$ test, ***$P < 0.001$, **$P < 0.01$ (95% confidence interval). Source data are provided as the Source Data file.

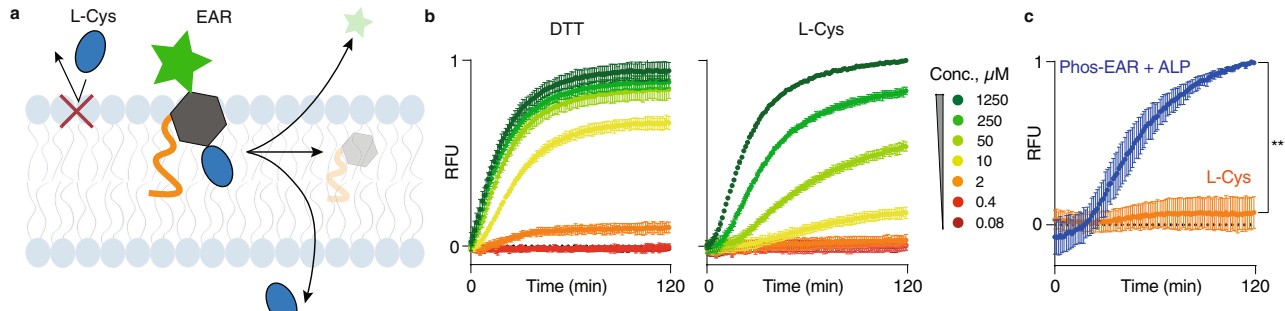

**Fig. 5 | Receptor-mediated vs diffusive signaling. a** Schematic illustration of diffusive signaling by L-Cys vs receptor-mediated signaling by Phos-EAR. **b, c** Fluorescence intensity evolution profiles in solutions of liposomes that contain the disulfide-based zymogen of papain and the protease substrate (Arg-AMC); zymogen activation is performed by DTT or L-Cys via diffusion across the lipid bilayer (**b**) or comparing 2 μM L-Cys to 2 μM Phos-EAR installed into the liposomes and triggered by the addition of ALP (**c**). **b, c** Results shown are based on three replicates with triplicates ($N = 3$, $n = 3$) presented as mean ± SD.; RFU = relative fluorescence expressed in arbitrary units. **c** Statistical significance at endpoint was calculated via an unpaired two-tailed $t$ test, ***$P < 0.001$ (95% confidence interval). Source data are provided as the Source Data file.

the phosphate trigger group, the ensuing decomposition of the self-immolative linker as a signal transduction mechanism, the release of the secondary messenger molecule L-Cys, and finally the activation of the zymogen into the catalytically active papain, which decomposes the fluorogenic substrate.

Signal transduction in liposomes equipped with Glu-EAR or Gus-EAR was also statistically significant (Fig. 4b). In itself, the observed activity for Glu-EAR is worthy of note because results in Fig. 3b revealed no enzymatic activity for GLU on the Glu-EAR precursor molecule **11**, which we attributed to inhibition of GLU by the aliphatic $C_{18}$ lipid chain. Results in Fig. 4 suggest that when associated with the liposomes, the lipid anchor of the Glu-EAR is not inhibiting catalysis by GLU to the same extent. Nevertheless, transmembrane signaling by Glu-EAR was rather inefficient compared to that by Gus-EAR. The key difference between glucose and glucuronic acid is that the former is a non-ionizable carbohydrate, whereas the latter has an ionizable carboxylic acid functionality. The ionization of the trigger group impacts its polarity, which likely translates into its propensity for an exofacial, aqueous placement at the liposome surface, and therefore the accessibility to the enzyme for the receptor activation. Transmembrane signaling with Gus-EAR upon activation with GUS was as efficacious as in the case of Phos-EAR, although it required a final bulk receptor concentration of at least 20 μM. Taken together, Fig. 4 illustrates that Phos-EAR stands out as being equally efficacious to Gus-EAR but

superior in working at significantly lower receptor content. For these reasons, further characterization of transmembrane signaling was carried out using the Phos-EAR molecule.

The next series of experiments were designed to validate that communication across the lipid bilayer in the EAR-equipped liposomes is a transmembrane signaling event, and not based on solute diffusion across the bilayer (Fig. 5a). First, we validated the sealed nature of the lipid bilayer within the liposomes. Towards this end, liposomes with the encapsulated papain zymogen and Arg-AMC were exposed to dithiothreitol (DTT), one of the most common biochemical reducing agents. DTT is well soluble in both organic and aqueous phase (calculated logP ~ 0) and is expected to diffuse freely through the lipid bilayer. Indeed, as little as 2 μM of DTT was sufficient to register an increase in fluorescence of the solution, indicating activation of papain from its disulfide-based chemical zymogen within the confines of the liposomes (Fig. 5b). In turn, the amino acid L-Cys is a zwitterion and is expected to have a lower permeability through the lipid bilayer. Indeed, activation potency of L-Cys was lower than for DTT and 2 μM concentration of L-Cys was insufficient to reactivate papain from the zymogen across the lipid bilayer. Nevertheless, with increased concentration of L-Cys, the gradually enhanced concentration gradient becomes sufficient to force permeation through the lipid bilayer, as evidenced by the emerged enzyme activity. Thus, the activation behavior for the encapsulated papain zymogen in response to the

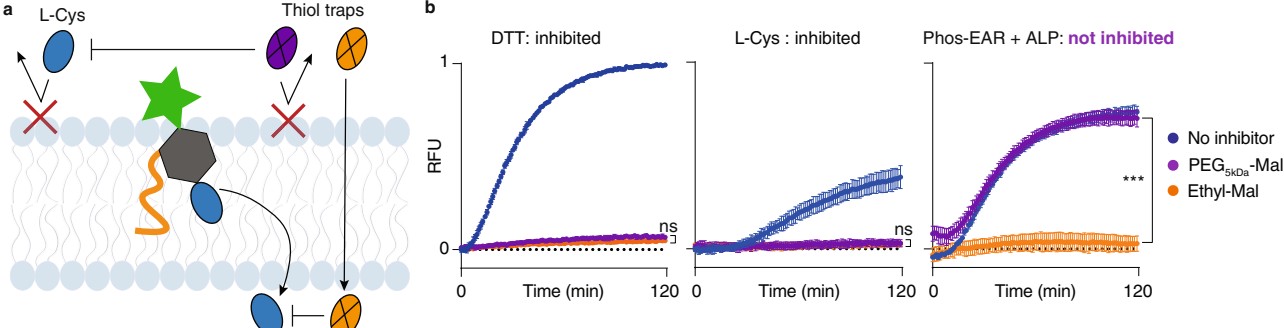

**Fig. 6 | Inhibition of signaling by thiol traps. a** Schematic illustration for thiol inhibition by N-ethylmaleimide (Ethyl-Mal) or PEG maleimide (PEG$_{5kDa}$-Mal) in the internal and external solution, the lipid bilayer, and/or within the liposome lumen. **b** Fluorescence intensity in solutions of liposomes that contain the papain zymogen and the protease substrate Arg-AMC; zymogen reactivation is performed by DTT, L-Cys, or Phos-EAR upon addition of ALP; zymogen reactivation is performed in the presence of N-ethylmaleimide (orange) or PEG$_{5kDa}$ maleimide (purple) or no maleimide (blue). Concentration of DTT, L-Cys, and Phos-EAR was 10 μM, and of the two maleimides it was 50 μM. Results shown are based on three replicates with triplicates ($N = 3$, $n = 3$), relative to the max of the DTT sample and shown as mean ± SD. RFU = relative fluorescence expressed in arbitrary units. Statistical significance was evaluated for the endpoint of the Ethyl-Mal and PEG$_{5kDa}$-Mal samples within each individual plot via an unpaired two-tailed $t$ test, ***$P < 0.001$ (95% confidence interval), ns = non-significant. Source data are provided as the Source Data file.

added DTT or L-Cys is consistent with the semi-permeable nature of the lipid bilayer.

We then compared activation of the encapsulated papain zymogen by L-Cys, added externally or released via the activation of Phos-EAR, at identical concentrations (Fig. 5c). In this experiment, matching efficacy of zymogen activation by the two treatments would indicate that the secondary messenger molecule is released from EAR into solution bulk, which contradicts the proposed transmembrane signaling. In contrast, superior efficacy of EAR over the externally administered L-Cys would indicate that the secondary messenger is released from EAR into the host liposome, as a transmembrane signaling event. In our experiments, at L-Cys concentration of 2 μM we observed negligible activation of the encapsulated zymogen with the amino acid (Fig. 5c). At the same concentration of liposomes and 2 μM content of Phos-EAR, the synthesized receptor molecule produced a strong signaling event, as evidenced by pronounced evolution of solution fluorescence in response to the added ALP enzyme. This result provides the strongest evidence to postulate receptor-mediated transmembrane signaling.

Lastly, we also aimed to gain a better insight into the EAR anchoring within the lipid bilayer. For this, we used two maleimide-based thiol trap molecules, namely N-ethylmaleimide and a macromolecular counterpart based on poly (ethylene glycol) (PEG) maleimide, molar mass 5 kDa. The former is membrane-permeable and is expected to react with thiols in the solution bulk, within the bilayer, and within the liposome lumen. In contrast, macromolecular PEG should experience restricted membrane permeability and its thiol inhibitory activity should be confined to the external thiols (Fig. 6a). Liposomes were prepared to contain papain in the form of its disulfide-based chemical zymogen and the protease substrate Arg-AMC. Thiol trap molecules were added to the liposomes along with DTT or L-Cys (10 μM) taken as external, diffusive zymogen activators. At the selected conditions, activity of the maleimide inhibitors was very fast and we observed only minor, if any, activation of the encapsulated zymogen in both cases (Fig. 6b).

In the Phos-EAR incubated liposomes triggered via the addition of the ALP, papain activation was suppressed only by the membrane-permeable ethylmaleimide. Receptor performance was not affected by the PEG-based inhibitor, indicating that the L-Cys in the structure of EAR is not accessible to the PEG maleimide for a chemical reaction (Fig. 6b). It also confirms that PEG is membrane-impermeable and does not inhibit the papain enzyme itself (once it is liberated from the corresponding zymogen). Together, the results in Figs. 4–6 illustrate

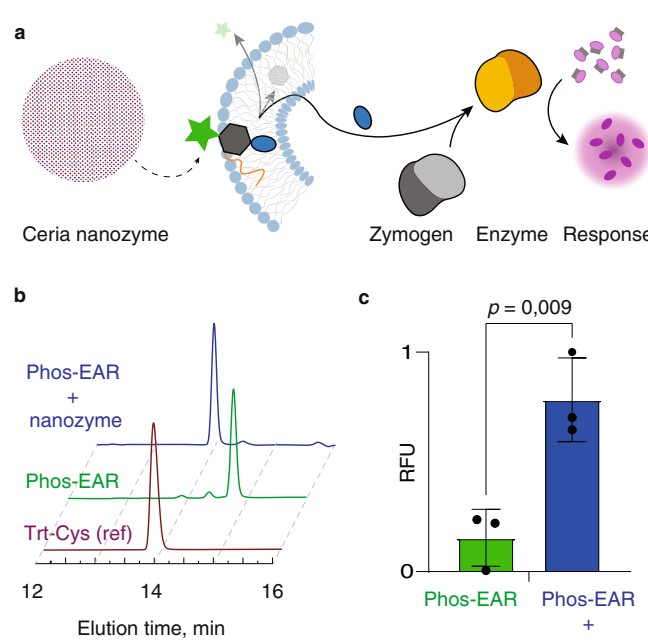

**Fig. 7 | Nanozyme triggered transmembrane signaling. a** Schematic illustration of transmembrane signaling mediated by Phos-EAR and triggered by a ceria nanozyme. **b** HPLC data illustrating successful release of S-Trt-Cys from the structure of the Phos-EAR precursor compound **8** by added ceria nanozyme. **c** Two hours endpoint fluorescence intensity measurement for solutions of liposomes that contain encapsulated papain zymogen and the protease substrate and Phos-EAR receptor, upon addition of ceria nanozyme. **c** Shown results are based on three replicates with triplicates ($N = 3$, $n = 3$) and shown as mean ± SD; RFU = relative fluorescence expressed in arbitrary units; statistical evaluation was performed via an unpaired two-tailed $t$ test. Raw data are provided as the Source Data file.

that the EAR molecules transmit the signal via a chemical signaling mechanism, not via a passive diffusion of a released activator.

### Receptor activation using a nanozyme

Another attractive feature of the Phos-EAR is that it accommodates signaling in response to non-biological activators such as nanozymes, non-biological (not protein-based) catalysts (Fig. 7a). Nanozymes are increasingly used in diverse areas of biomedicine and biotechnology, specifically in biosensing[41]. In our own recent work, we designed nanozymes to achieve conversion of prodrugs into the corresponding

therapeutic molecules for anticancer or antibacterial action[38,42]. In one specific example, we designed ceria nanoparticles as mimics to phosphatases with activity on a range of biological (pyro)phosphates and chemical prodrugs[38]. Herein, we exposed the Phos-EAR precursor **8** and observed that the ceria nanozyme effectively released *S*-Trt-Cys from this molecule, in a manner fully mimicking activity of ALP (Fig. 7b). Further, we incubated ceria nanoparticles with the Phos-EAR-equipped liposomes that contained the papain zymogen and protease substrate Arg-AMC. We observed pronounced increase in fluorescence after two hours, indicating successful receptor signaling in a syncell in response to a nanozyme (Fig. 7c). In this case, the syncell responded to the presence of a nanozyme by an intracellular enzymatic reaction, resulting from the signal transduction across the lipid bilayer accomplished using a synthetic, chemical receptor.

Taken together, results of this study present the design and performance of chemical, artificial receptors that perform signaling across sealed biological membranes. The similarity of our synthetic receptors with biological and chimeric counterparts is in having a well-defined exofacial part, having an in-membrane signal transducer, in responding to a biological primary messenger, and in activation of an enzyme for downstream signaling. The main difference, and therefore the main innovative aspect of the EAR signaling, is in the mechanism of signal transduction: the 1,6-benzyl elimination of a self-immolative linker. Signal transduction design presented herein extends our work on the protein cysteinome[37] and the released secondary messenger is an amino acid L-Cys, for activation of the disulfide-based chemical zymogens. Thiol inhibition and diffusion studies validated the receptor-like mechanism of signal transduction mediated by the EAR molecules.

We develop artificial receptors specifically for the use in syncells. Receptors are indispensable to establish communication between cells in a multicellular ensemble. However, to our knowledge, signaling in syncells based on natural receptors has never been accomplished, because receptor homeostasis is maintained via complex biochemical pathways. While this research goal has its merits, we chose to perform de novo engineering of a chemical mechanism of signal transduction. For signal transduction, we use a self-immolative molecule, which affords a triggered, traceless release of a secondary messenger for activation of an encapsulated zymogen. In the current form, the EAR molecules are activated by an enzymatic reaction, which is different to the natural sensing of soluble ligands via non-covalent interactions. However, enzymatic activation offers unique opportunities to establish communication between syncells within their ensemble or with natural counterpart. This is because (i) mammalian cells and tissue often secrete enzymes and the enzymatic fingerprint can indicate an onset of a disease;[43] and (ii) enzymatic activation of receptors in syncells will be orthogonal to the natural signaling pathways. From a different perspective, a highly promising emerging application of syncells is within the field of biosensing[10]. Receptor-mediated activation of an encapsulated enzyme offers innovative applications because within a syncell the encapsulated protein is protected from the membrane-impermeable inhibitors. We believe that the results of our work will empower the field and future of the engineering of synthetic cells[13,19,44,45].

## Methods

For details on compound synthesis and characterization, see Supplementary Methods.

### General information on biological experiments

All chemicals were purchased from Sigma Aldrich/Merck, unless other is stated. The buffer used for all experiments was a 20 mM (4-(2-hydroxyethyl)−1-piperazineethanesulfonic acid buffer, pH 6.8, filtered through a 0.2 μm filter. It will be referred to as HEPES buffer in the following. Dynamic light scattering (DLS) experiments were performed using a Malvern Zetasizer Nano S90 on all liposome samples to ensure successful extrusion (acceptable size and dispersity) with and without the presence of the receptor molecules. Enzymatic catalysis kinetic experiments were performed using a BioTek Synergy H1 microplate reader with Gen5 3.5 software. 7-Amido-4-methylcoumarin (AMC) fluorescence was measured at ex.: 348 nm, em.: 443 nm. Confocal laser scanning microscopy was performed on a Zeiss Aixo Observer LSM 700.

### Papain zymogen activity

Papain was purchased from Sigma Aldrich as an inactive enzyme (disulfide-containing zymogen). As quality control the inactivity was tested by measuring papain activity in solution with and without the addition of the reducing agent dithiothreitol (DTT). Reagent concentrations: 0.03 g/L Papain, 5 μM $N_\alpha$-benzoyl-L-arginine-7-amido-4-methylcoumarin (Arg-AMC) from Bachem, 10 μM DTT. In supplemental Fig. 2 results are presented as mean ± SD of three independent experiments with triplicates ($N = 3$, $n = 3$) confirming inactivity of the Papain and subsequent activation when DTT is present.

### Visualization of EAR in GUVs

In total, 0.25 μg of lipids containing EPC and cholesterol at molar ratio of 60:40 was made into a thin film on top of an indium tin oxide (ITO) glass and left at vacuum overnight. Afterward samples were hydrated with 300 μL of buffer containing 300 mM sucrose in phosphate-buffered saline (PBS). Then GUVs were formed by on a Nanion Technologies Vesicle Prep Pro, Freq: 10.0 Hz, amplitude: 3.0 V, temperature 37 °C, for 3 h. GUVs were stored at 4 °C and used within 2–3 days after formation. Suspension of GUVs was prepared in a solution containing 300 mM glucose in PBS and was charged with Gus-EAR and the thiol-reactive fluorescein maleimide. A sample was subsequently treated for 1 h with GUS at 150 mg/L. Control samples without treatment with GUS, bearing only Gus-EAR or treated only with fluorescein maleimide were also prepared. Vesicles were analyzed by confocal laser scanning microscopy. Imaging settings were adjusted and selected with the EAR + /Mal+ sample and kept constant for the remaining samples.

### Liposome preparation

Alfa-phosphatidylcholine from egg yolk (EPC) was dissolved in chloroform (25 g/L). 250 μL (6.25 mg) of this solution was added to a 5 mL pear-shaped glass flask. A lipid film was prepared in the flask by evaporation of chloroform using nitrogen flow while rotating the flask. The lipid film was further dried under vacuum overnight. The hydration solution (10 g/L papain zymogen and 1 mM Arg-AMC substrate in HEPES buffer, 100 μL) was added to the lipid film and vortexed until the film was fully hydrated. After successful hydration of the lipids, additional buffer was added to a final volume of 250 μL. The liposome mixture was then extruded 15 times through a 200-nm filter (Whatman® Nuclepore™ Track-Etched Membranes) before the liposomes were purified using size exclusion chromatography (SEC). This was done by packing a column (H: 10 cm, D: 1 cm) with Sepharose® CL-2B. The column was washed with HEPES buffer three times volume of the packed column, before the liposome sample was applied. Elution was conducted using HEPES buffer, liposome-containing fractions were identified as turbid fractions and combined. For larger experiments, the preparation of liposomes was scaled to suit the needed amount of liposome solution.

### DLS measurements

All liposome samples were analyzed using DLS and only used when the size were within the range of 200–250 nm and with a polydispersity index below 0.2. All measurements were conducted as three independent measurements. Supplemental Fig. 3 illustrates representative DLS measurements of 200 nm liposomes with and without EAR at the concentrations used in the different experiments.

## Determination of EAR stock concentration

To determine the concentrations of the final EAR stocks dissolved in DMSO, the sulfhydryl concentration was determined using a 5,5′-dithio-bis-(2-nitrobenzoic acid) (DTNB) standard curve. The standard curve was prepared by diluting DTNB in 50 mM HEPES buffer pH 8, at a final concentration of 1 mM. L-Cys of varying concentrations (0, 0.03125, 0.0625, 0.125, 0.25, 0.5, and 0.75 mM) was added from a DMSO stock to the solution containing DTNB. The reactions were left for 10 min after which the absorbance of each sample was measured by UV–Vis spectrometry at a wavelength of 412 nm. The experiments were carried out in three independent replicates with triplicates ($N = 3$, $n = 3$). Supplemental Fig. 4 shows the results of the calibration curve. The DMSO stocks containing the different EAR were diluted 100-fold upon addition to the DTNB solution. The mixture was left to react for 10 min, and the absorption was measured using UV–Vis at 412 nm. The concentration of EAR was determined using the standard curve slope from the absorption results of the L-Cys-containing samples.

## Incorporation of EAR into liposomes

The liposomes were prepared as described above. The different EAR compounds (Phos-EAR stock: 10 mM, Glu-EAR stock: 4.4 mM, Gus-EAR stock 3.9 mM in DMSO) was added directly to the solution containing purified liposomes (vol % of DMSO = 1) and left for 30 min at room temperature, letting the receptor anchor in the liposome membrane. Control samples without EAR were prepared by the addition of equivalent amount of DMSO to the liposomes. The stated concentration of EAR is the final concentration within the 96-well plate with a final volume of 100 μL in each well.

## Receptor-mediated signaling in liposomes

Phos-EAR-containing liposomes were prepared as described above. The final concentration of Phos-EAR in the liposome-containing solution was 2 μM. Receptor-mediated signaling was evaluated as a kinetic study, where the liposomes were treated externally with the activating enzyme alkaline phosphatase (ALP); enzymatic activity of the liposome-encapsulated papain was recorded as a fluorescent output due to turnover of Arg-AMC, using a plate reader. The samples were prepared in a 96-well plate and consisted of 50 μL liposomes (with or without EAR), 10 μL enzyme (150 mg/L) and buffer to a final volume of 100 μL. The control without enzyme was treated with 10 μL of HEPES buffer. The enzyme was added as the last component using a multichannel pipette and the plate was directly subjected to the kinetic measurements. The experiment was reproduced in three independent experiments, each containing three technical replicates for each experimental condition ($N = 3$, $n = 3$) within the same preparation of liposomes. The background fluorescence of the liposome control sample with or without ALP was subtracted from the data; the data were normalized to the maximum recorded fluorescence (the Phos-EAR + ALP sample) within each independent replicate and plotted as the mean ± SD.

Receptor signaling by Glu-EAR and Gus-EAR was recorded through the same protocol with minor alterations. Specifically, the final concentration of Glu-EAR and Gus-EAR was 20 μM, receptor activation was performed with β-glucosidase (GLU) or β-glucuronidase (GUS), respectively. The data were normalized to the maximum recorded fluorescence (the Gus-EAR + GUS sample) within each independent replicate and plotted as the mean ± SD.

## Diffusive papain activation by L-Cys and DTT in liposomes

Liposomes containing encapsulated papain zymogen and Arg-AMC were prepared as described above. The following were combined in a 96-well plate: 50 μL liposomes, 10 μL L-Cys or DTT from a dilution series (final concentration range: 1250–0.08 μM) and HEPES buffer to a final volume of 100 μL. Background control samples without liposomes were also included. The L-Cys and DTT solutions were added using a multichannel pipette as the last added component right before the plate was subjected to the kinetic measurements. Enzymatic activity of papain was recorded as the evolution of fluorescence signal due to the turnover of Arg-AMC, using a plate reader. The experiment was reproduced in three independent replicates with triplicates ($N = 3$, $n = 3$) within the same preparation of liposomes. The raw experimental data were normalized to the maximum fluorescence intensity value within each independent replicate and plotted as the mean ± SD.

## Transmembrane signaling by Phos-EAR compared to diffusion-based signaling by L-Cys

Liposomes were prepared as described above and contained encapsulated papain zymogen and Arg-AMC substrate, with or without incorporation of Phos-EAR. In the wells of a 96-well plate, 50 μL liposomes were combined with 40 μL of HEPES buffer. To the solution of Phos-EAR containing liposomes was added 10 μL ALP (150 mg/L); to the solution of liposomes without Phos-EAR was added 10 μL L-Cys giving a final concentration of 2 μM. The controls without ALP and L-Cys received 10 μL of extra buffer volume. The L-Cys and ALP solutions were added using a multichannel pipette as the last components. The activity of the encapsulated papain was recorded as the evolution of fluorescence due to enzymatic turnover of Arg-AMC, using a plate reader. The experiment was reproduced in three independent experiments each conducted in triplicates ($N = 3$, $n = 3$) within the same preparation of liposomes. From the experimental data, the appropriate background was subtracted and normalized to the maximum fluorescence signal of the Phos-EAR + ALP sample within each independent replicate and plotted as the mean ± SD.

## Inhibitor (thiol trap) experiment

Liposomes were prepared as described above and contained encapsulated papain zymogen, Arg-AMC substrate, and optionally Phos-EAR for transmembrane signaling. N-Ethylmaleimide or MeO-PEG$_{5KDa}$-Maleimide (IRIS Biotech) were used as thiol traps. In the wells of a 96-well plate as follows were combined 10 μL thiol trap (final concentration 50 μM) and 50 μL liposomes (with Phos-EAR or without), and 30 μL of HEPES buffer. Solutions of the Phos-EAR containing liposomes received 10 μL ALP enzyme for the Phos-EAR samples (150 mg/L); solutions of liposomes without the added receptor received 10 μL L-Cys or DTT (final concentration 10 μM). The controls without ALP, L-Cys, and DTT received 10 μL of extra buffer. The L-Cys, DTT and ALP were added using a multichannel pipette as the last added volumes. Activation of the encapsulated papain was recorded as the evolution of fluorescence due to enzymatic turnover of Arg-AMC. The experiment was reproduced in three independent replicates, each containing three technical replicates ($N = 3$, $n = 3$) within the same preparation of liposomes. The raw data were normalized to the maximum fluorescence signal achieved for papain activation using DTT in the absence of added inhibitors, and presented as mean ± SD.

## CeO$_2$ (nanozyme) mediated Phos-EAR activation in liposomes

Liposomes were prepared as described above and contained encapsulated papain zymogen, Arg-AMC substrate, and Phos-EAR receptor. The final concentration of Phos-EAR in the liposome samples was 10 μM. The samples were prepared in a 96-well plate through the mixing of 50 μL liposomes (with or without Phos-EAR), 10 μL ceria nanozyme (1 g/L) and buffer to a final volume of 100 μL. The control without nanozyme received 10 μL of extra buffer. The nanozyme was added as the last added volume. The plate was incubated with shaking for 2 h at 37 °C before an endpoint measurement of AMC fluorescence was measured using a plate reader. The experiment was reproduced in three independent experiments ($N = 3$) each containing three technical replicates ($n = 3$) within the same preparation of liposomes. The data are presented relative to the maximum recorded fluorescence signal as the mean ± SD.

## Statistical analysis

All statistical analyses are based on three independent experiments with three technical replicates ($N = 3$, $n = 3$) and are presented as the mean ± SD. The statistical significance was established via an unpaired, two-tailed $t$ test comparing the endpoint measurements using the software Graphpad Prism 9.0.

## Reporting summary

Further information on research design is available in the Nature Portfolio Reporting Summary linked to this article.

## Data availability

All the data generated in this study are provided in the Source data file or available from the corresponding author on request. Source data are provided with this paper.

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

## Acknowledgements

We wish to acknowledge funding from the Lundbeck Foundation (Grant No R164-2013-15291), the Carlsberg Foundation (Grant No. CF19-0275), the Independent Research Fund Denmark (DFF FNU Grant No 0135-00162B), and the Novo Nordisk Foundation (Grant No NNF20OC0062131), all to A.N.Z.

## Author contributions

Conceptualization: A.N.Z., A.B.S., R.W., A.B.P., K.B.L., and P.M.M.; investigation: A.B.S., A.B.P., K.B.L., P.M.M., J.H.J., L.D., L.F.N., and S.S.; formal analysis: A.B.S., A.N.Z., K.B.L., and P.M.M.; writing: A.N.Z. and A.B.S.; funding acquisition, project administration, and supervision: A.N.Z.

## Competing interests

The authors declare no competing interests.
