## [Peer Review File · Nature Communications]

REVIEWER COMMENTS

Reviewer #1 (Remarks to the Author):

In this paper the authors engineer a synthetic signal transduction system operating in liposomes and transducing a biological signal from outside to the inside. The problem is highly relevant and important for the field. How to transmit detection of an external signal to the interior of a vesicle would have profound impacts on several fields, including biosensor engineering, if a system modular and scalable enough was to be designed.

Here the authors designed a clever system, using a self-immolating system kept inactive, the GUS enzyme cleaves a glucuronide moiety on the Exo surface of the liposome, the self-immolating linker is activated, cleaves the receptor, and releases a messenger (L-cysteine) in the interior of the vesicle which reacts with encapsulated enzymes to provide a signal. The data are repeated in independent experiments and the results are clear.

This is an important step the system is clever. However, to me there are limitations in terms of demonstrated sensing modalities and modularity that preclude publication in this form Nat Comms:

1. Mostly, the biological signal is in fact an enzymatic (cleavage) activity. While this is a valid demo, many receptors detect ligands, and showing this capability would be of great interest, albeit harder to realize. Other enzymes should work in theory but only one is shown here. So indeed the system detects a "biological signal", but in terms of application, without detecting a soluble ligand the application is more limited (although some important stuff could be done, such as detecting proteases for diagnostics). The authors should be clear on the type of biological signal that can be detected or not.

2. Even with a specific type of biological signal, demonstration of scalability is important. Can the system re modified to detect other enzymes? This would be a key point to show in my opinion.

If these points were satisfied, the paper would represent a sufficient advance for publication in Nat comms.

The authors should in addition discuss the imitations of their system how they would modify it to detect soluble ligands.

In addition, beside the conceptual advance of the system, why it would be interesting to use vesicles instead of bulk assay for enzyme activities assays should be discussed.

Other comments:

- The title is not precisely discussing the results and should be modified.
- The term Bionic sounds a bit strange and could be misleading.
- there is a lack of description of the potential applications
- In figure 1, the graphic describing the receptor operation should be made bigger and clearer, like in fig 4.
- Figures should show the vesicles in bright field too most important when no fluorescence is observed.
- Response to GUS could be plotted as a dose-response curve, e.g. taking values at a plateau as a function of GUS concentration. this would give a better idea of the response of the system. However, more intermediate points might be needed.
- plotting the data as μM equivalent FITC using a calibration FITC curve would help.
- The paper completely lacks any sort of discussion about the limitations of the system, limited sensing modalities, but also irreversibility, kinetics, the limit of detection (How fast is it compared to bulk reaction, how sensitive etc...).
- The paper lacks a description of possible applications of such systems, either for basic research or engineering (e.g. biosensors)

In general, the paper text should be expanded, there is much space for that.

Reviewer #2 (Remarks to the Author):

In this contribution a self-immolative strategy is described to release an enzyme activating component in GUVs. The release is triggered by the addition of an enzyme in the external environment. The authors have demonstrated that association of the masked component with the GUVs improves substantially their ability to release the thiol in the lumen of the vesicles. The authors claim that this is a biomimetic concept mimicking signal transduction by living cells. This latter part can be argued. The component doesn't span the membrane, and there is no regulatory

mechanism involved. Removal of the enzyme from the external environment does not reset the system. The component is furthermore consumed during the process. These aspects are all different from regular signaling pathways found in nature. This makes this contribution an interesting technical advance, but from a conceptual point of view less groundbreaking as claimed in the manuscript. Furthermore, there is already quite some literature on chemical communication/signaling across membranes. The work of the Devaraj and Mann groups for example should be mentioned, and also the IVTT systems described by Adamala and Manssy deserve attention. This part of the literature is insufficiently covered.

Technical comments:

The authors observe in Fig 2 a drastic decrease in fluorescence when cysteine is released by GUS and the reaction with maleimide is prevented. However, GUS should not be able to reach the internal leaflet of the bilayer. As there doesn't seem to be a preference of the amphiphile to be on either side of the bilayer, how do the authors explain the strong decrease in signal in Fig 2? One would expect that 50% of the amphiphile cannot be cleaved.

I don't understand the explanation why an increase in EAR concentration to 250 μ M leads to a decrease in signal. If membrane saturation occurs then the rate should stay at least the same as for the 50 micromolar concentration conditions. The authors should provide a better explanation of this observation.

As EAR is not a membrane spanning molecule, cysteine release will happen both internally and externally. The authors should determine the concentration of released cysteine in the bulk phase.

Reviewer #3 (Remarks to the Author):

This work describes a synthetic system imitating features of signal transduction. The so-called artificial signal cascade is achieved using a self-immolative linker (SIL) embedded in a lipid bilayer. Cleavage of a glucuronic ester in SIL by a GUS enzyme leads to liberation of cysteine and papain activation.

This work is described in the framework of supramolecular models, such as models based on adaptive dynamic networks, which elucidate (and one day perhaps mimic) the underlying rules of signal transduction across membrane in nature. Such systems usually join supramolecular recognition on liposomes to transmembrane signaling across lipid bilayers.

Although this is definitely an interesting study, the relevance of this work to bionic receptors and artificial signal transduction mechanisms is unclear. The authors claim that difference with the natural receptors is that signal transduction across the lipid bilayer was performed using the tools of organic chemistry. However, the entire signaling process discussed in this study is based on

enzymes. In fact, the artificial receptor is a kind of prodrug that is enzymatically cleaved. Incorporating prodrugs/substrates in liposome membranes to initiate artificial signal cascades is an interesting development; however, unless the author demonstrate some specific applications of the system, or entirely synthetic signal cascades, it should be published in a specialized journal.

REVIEWER COMMENTS

Reviewer #1 (Remarks to the Author):

Comment 1 (C1). In this paper the authors engineer a synthetic signal transduction system operating in liposomes and transducing a biological signal from outside to the inside. The problem is highly relevant and important for the field. How to transmit detection of an external signal to the interior of a vesicle would have profound impacts on several fields, including biosensor engineering, if a system modular and scalable enough was to be designed.

Here the authors designed a clever system, using a self-immolating system kept inactive, the GUS enzyme cleaves a glucuronide moiety on the Exo surface of the liposome, the self-immolating linker is activated, cleaves the receptor, and releases a messenger (L-cysteine) in the interior of the vesicle which reacts with encapsulated enzymes to provide a signal. The data are repeated in independent experiments and the results are clear.

Response 1 (R1): We thank the reviewer for highlighting that problem is highly relevant and important, and that our results could have a profound impact on the field if the system was proven to be modular and scalable. We have invested significant time to address these points and **now show that the system as modular and scalable.**

Specifically, we re-designed the receptor synthesis, which is now conducted via a divergent scheme; from the key modular receptor scaffold we are able to synthesize receptors that respond to diverse stimuli. The receptor scaffold is synthesized on gram scale, thus illustrating that this synthesis has been scaled up (see **new Figures 2,3**).

C2: This is an important step the system is clever. However, to me there are limitations in terms of demonstrated sensing modalities and modularity that preclude publication in this form Nat Comms: Mostly, the biological signal is in fact an enzymatic (cleavage) activity. While this is a valid demo, many receptors detect ligands, and showing this capability would be of great interest, albeit harder to realize.

R2: We fully agree that natural receptors typically detect soluble ligands via non-covalent interactions. However, we emphasize that we engineer artificial receptors for synthetic cells.

We address the fundamental challenge of performing transmembrane signalling. This reviewer highlights this challenge of “transducing a biological signal from outside to the inside <is> highly relevant and important for the field.” Chemical receptors do not have to be engineered the same way as in nature, and signalling modalities do not have to be identical to those employed in nature. In fact, the signalling mechanism *has* to be de novo engineered because natural receptors (e.g. GPCR) rely on multi-enzymatic pathways of regulation and re-assembly and can hardly be maintained in a synthetic cell.

C3. Other enzymes should work in theory **but only one is shown here.**

R3: We have specifically addressed this point and show that the receptor scaffold is modular: we engineered three independent receptors, establishing response to three activating enzyme AND a nanozyme (an inorganic mimic of an enzyme), see **new Figures 3, 4, 7.**

C4: So indeed the system detects a “biological signal”, but in terms of application, without detecting a soluble ligand the **application** is more limited (although some important stuff could be done, such as detecting proteases for diagnostics). The authors should be clear on the type of biological signal that can be detected or not.

R4: To provide clarity, we added a paragraph to the Introduction of the manuscript and specified that our envisioned application is engineering of synthetic cells. We also added a discussion at the end of the manuscript to spell which signals can be detected by the designed receptors, and a justification of utility for the enzymatic activators.

C5: Even with a specific type of biological signal, demonstration of scalability is important. Can the system be modified to detect other enzymes? **This would be a key point to show in my opinion.**

R5: We have specifically addressed this comment and designed three independent receptors, that detect different enzymes. In doing so, we purposefully address this key point expressed by this reviewer. Please see **new Figures 3, 4, 7.**

C6: If these points were satisfied, the paper would represent a sufficient advance for publication in Nat comms.

R6: We sincerely hope that we addressed the points raised by this reviewer and that the manuscript is sufficiently improved, to merit publication in Nature Communications.

C7: The authors should in addition discuss the limitations of their system how they would modify it to detect soluble ligands. In addition, beside the conceptual advance of the system, why it would be interesting to use vesicles instead of bulk assay for enzyme activities assays should be discussed.

R7: We agree that these points are important to discuss; a short discussion has been added to the manuscript. Specifically, new Figure 6 illustrates that receptor mediated enzymatic read-out can be performed in the presence of the membrane impermeable enzyme inhibitors, which opens up opportunities for biosensing applications.

Other comments:

C8: The title is not precisely discussing the results and should be modified.

R8: The content of the manuscript has changed a lot during revision and indeed we found it appropriate to change the title.

C9: The term Bionic sounds a bit strange and could be misleading.

R9: we changed the term “bionic” to “synthetic”.

C10: there is a lack of description of the potential applications

R10: In our opinion, the primary value of our work lies in the realm of fundamental science, not technology. Nevertheless, recent developments illustrate that artificial cells become important in applications such as biotechnology and biosensing. We believe that synthetic receptors make artificial cells stimuli-responsive, which is arguably the most important feature of a biosensor. We added these points to the discussion in the manuscript (page 2, 10-11).

C11. In figure 1, the graphic describing the receptor operation should be made bigger and clearer, like in fig 4.

R11: done

C12: Figures should show the vesicles in bright field too most important when no fluorescence is observed.

R12 : Figure 4A (visualization of GUVs) was modified to independently label the lipids (which we found more illustrative than bright field imaging of GUVs)

C13 - Response to GUS could be plotted as a dose-response curve, e.g. taking values at a plateau as a function of GUS concentration. this would give a better idea of the response of the system. However, more intermediate points might be needed.

R13 : This figure was removed from the manuscript during revision.

C14: plotting the data as μM equivalent FITC using a calibration FITC curve would help.

R14 : this figure was removed during the manuscript revision.

C15 - The paper completely lacks any sort of discussion about the limitations of the system, limited sensing modalities, but also irreversibility, kinetics, the limit of detection (How fast is it compared to bulk reaction, how sensitive etc...).

- The paper lacks a description of possible applications of such systems, either for basic research or engineering (e.g. biosensors)

In general, the paper text should be expanded, there is much space for that.

R15 : We have significantly expanded the discussion section to specifically address the points raised by the reviewer (pages 2, 10-11).

Reviewer #2 (Remarks to the Author):

Comment 1. In this contribution a self-immolative strategy is described to release an enzyme activating component in GUVs. The release is triggered by the addition of an enzyme in the external environment. The authors have demonstrated that association of the masked component with the GUVs improves substantially their ability to release the thiol in the lumen of the vesicles. The authors claim that this is a biomimetic concept mimicking signal transduction by living cells. This latter part can be argued. The component doesn't span the membrane, and there is no regulatory mechanism involved. Removal of the enzyme from the external environment does not reset the system. The component is furthermore consumed during the process. These aspects are all different from regular signaling pathways found in nature. This makes this contribution an interesting technical advance, but from a conceptual point of view less groundbreaking as claimed in the manuscript.

Response 1: We thank the reviewer for an in-depth evaluation of our work and for offering an insightful comparison with the natural receptors.

We note that receptors in nature are many by type and for example the NOTCH receptors are consumed during signaling and are not reset if the signaling ligand is removed (NOTCH needs to be re-synthesized).

Most importantly, we believe that the prime novelty of our work is in addressing the fundamental challenge of transmembrane signalling and as such we do not claim that our approach is biomimetic. We cannot help comparing our receptors to the natural counterpart, but only to bring our achievements into context, not to claim biomimicry. Our main challenge is the design of synthetic receptors for artificial cells. We added this justification to the revised manuscript (page 2).

C2. Furthermore, there is already quite some literature on chemical communication/signaling across membranes. The work of the Devaraj and Mann groups for example should be mentioned, and also the IVTT systems described by Adamala and Manssy deserve attention. This part of the literature is insufficiently covered.

R2: We regret that prior art was not referenced properly; the revised manuscript contains a full paragraph on the design of artificial cells, which includes references to the groups mentioned by the reviewer (page 2-3). These laboratories are without doubt at the forefront of this research. These reports are highly important in their own right.

However, the overall majority of the prior art on communication systems relied on signaling through diffusion of solutes as large as proteins and DNA, which is un-natural. Biological membranes are sealed and do not allow uncontrolled diffusion of proteins and DNA. We perform signaling through sealed biomimetic membranes, and this is our main accomplishment. We believe that compared to prior art, we address a stand-alone challenge that has been neglected by the field of biomimicry.

Technical comments:

C3: The authors observe in Fig 2 a drastic decrease in fluorescence when cysteine is released by GUS and the reaction with maleimide is prevented. However, GUS should not be able to reach the internal leaflet of the bilayer. As there doesn't seem to be a preference of the amphiphile to be on either side

of the bilayer, how do the authors explain the strong decrease in signal in Fig 2? One would expect that 50% of the amphiphile cannot be cleaved.

R3: during manuscript revision, this figure has been modified and does not contain the image discussed by the reviewer.

Q4 I don't understand the explanation why an increase in EAR concentration to 250 μM leads to a decrease in signal. If membrane saturation occurs then the rate should stay at least the same as for the 50 micromolar concentration conditions. The authors should provide a better explanation of this observation.

R4 : During manuscript revision, we have now achieved a reliable, efficacious, statistically significant receptor signaling with EAR concentrations as low as 2 μM and made no attempt to reproduce signaling at a 100-fold higher EAR content (to not waste the receptor stocks) and this figure was removed from the manuscript.

Q5: As EAR is not a membrane spanning molecule, cysteine release will happen both internally and externally. The authors should determine the concentration of released cysteine in the bulk phase.

R5: It is a highly relevant question, but we draw attention of the reviewer that tested side by side, at matched concentrations, 2 μM L-Cys shows negligible protein activation, whereas 2 μM EAR is highly efficacious. In practice, this means that concentration of L-Cys in the bulk phase will be between 0 and 2 μM , that is, inactive in our transmembrane signaling assay. Moreover, detection of thiols (e.g. via the Ellman's test) is unreliable in the (sub)micromolar concentration range.

Reviewer #3 (Remarks to the Author):

Comment 1. This work describes a synthetic system imitating features of signal transduction. The so-called artificial signal cascade is achieved using a self-immolative linker (SIL) embedded in a lipid bilayer. Cleavage of a glucuronic ester in SIL by a GUS enzyme leads to liberation of cysteine and papain activation.

This work is described in the framework of supramolecular models, such as models based on adaptive dynamic networks, which elucidate (and one day perhaps mimic) the underlying rules of signal transduction across membrane in nature. Such systems usually join supramolecular recognition on liposomes to transmembrane signaling across lipid bilayers.

Although this is definitely an interesting study, the relevance of this work to bionic receptors and artificial signal transduction mechanisms is unclear. The authors claim that difference with the natural receptors is that signal transduction across the lipid bilayer was performed using the tools of organic chemistry. However, the entire signaling process discussed in this study is based on enzymes. In fact, the artificial receptor is a kind of prodrug that is enzymatically cleaved.

Response 1: we thank the reviewer for highlighting that this work is interesting.

The signal transduction mechanism developed in our work is performed using tools of chemistry: it is based on the chemistry of self-immolative linkers.

Prior reports in the field (see Refs 22-28) have documented entirely chemical signaling cascades. The unfulfilled, higher challenge is to connect biology to biology using tools of chemistry. This challenge is specifically important to establish connection between synthetic cells and mammalian cells in their co-culture. It is this challenge that we addressed in our work.

We have added this short discussion to the revised manuscript (page 2-3)

Comment 2. Incorporating prodrugs/substrates in liposome membranes to initiate artificial signal cascades is an interesting development; however, unless the author demonstrate some specific applications of the system, **or** entirely synthetic signal cascades, it should be published in a specialized journal.

Response 2:

We see the main value of our work to lie in the realm of fundamental science: we work towards the design of artificial (synthetic) cells. Signal transduction is what makes cells responsive – and we take a major step forward in engineering such responsive behaviour into artificial cells. This is one application for the transmembrane signaling developed in this work.

The second application, with importance for biotechnology and biosensing, is that we demonstrate on-demand activation of catalysis in the presence of enzyme inhibitors (**new Figure 6**). This will be of the highest potential interest for biosensing applications (see Ref 10).

We added these points as discussion to the manuscript (page 2, 10-11)

Lastly, specifically to address the request from this reviewer, we added an illustration of a synthetic transmembrane signaling (**new Figure 7**): we established receptor activation using a nanozyme, an inorganic mimic of phosphatases; ensuing signal transduction was performed via 1,6-benzyl elimination. In this example, catalytic (enzymatic) read-out is used solely for signal amplification.

REVIEWERS' COMMENTS

Reviewer #2 (Remarks to the Author):

The authors have provided additional experiments and explanations regarding their paper. They have convincingly clarified a number of issues. For example, they have shown that different enzymes can be used as trigger (albeit with different efficiencies). Still, my basic opinion that this is a technological rather than a conceptual advance has not changed.

Furthermore, the statement that communication only happens with large molecules is not correct. Different authors have for example used the activation by small molecules that can diffuse freely across a closed membrane.

I also don't understand the response to my comment to the original fig 2. What does the modification entail?

Reviewer #3 (Remarks to the Author):

The revised communication more clearly describes the novelty of the work and the new experiments strengthen some points that were not supported in the original manuscript. Overall, the modified work becomes relevant to readers of this journal.

REVIEWER COMMENTS

Reviewer #2 (Remarks to the Author):

The authors have provided additional experiments and explanations regarding their paper. They have convincingly clarified a number of issues. For example, they have shown that different enzymes can be used as trigger (albeit with different efficiencies).

Response : we thank the reviewer for highlighting that the revised manuscript convincingly clarified a number of issues raised by the reviewers.

Comment: Still, my basic opinion that this is a technological rather than a conceptual advance has not changed. Furthermore, the statement that communication only happens with large molecules is not correct. Different authors have for example used the activation by small molecules that can diffuse freely across a closed membrane.

Response : the reviewer pinpoints here the difference between prior reports and our work : free diffusion of small molecules (and even more so the diffusion of larger molecules) through a membrane by definition means that this membrane is NOT closed, but open to this molecule. In our design of artificial cells, we aimed to mimic nature, to capitalize on the benefits of cellularity and compartmentalization whereby solutes cannot diffuse freely and whereby receptor-mediated process become pivotal.

Comment: I also don't understand the response to my comment to the original fig 2. What does the modification entail?

(Original question : The authors observe in Fig 2 a drastic decrease in fluorescence when cysteine is released by GUS and the reaction with maleimide is prevented. However, GUS should not be able to reach the internal leaflet of the bilayer. As there doesn't seem to be a preference of the amphiphile to be on either side of the bilayer, how do the authors explain the strong decrease in signal in Fig 2? One would expect that 50% of the amphiphile cannot be cleaved.

Original answer : during manuscript revision, this figure has been modified and does not contain the image discussed by the reviewer.)

Response : The reviewer possibly overlooks that the receptor insertion is achieved in our work into the pre-formed lipid bilayer, from a solution, via spontaneous insertion of the hydrophobic part of the receptor molecule into the bilayer. This means that all receptor molecules are inserted only to one side of the bilayer, the external leaflet.

Reviewer #3 (Remarks to the Author):

The revised communication more clearly describes the novelty of the work and the new experiments strengthen some points that were not supported in the original manuscript. Overall, the modified work becomes relevant to readers of this journal.

Response : We sincerely thank the reviewer for this positive evaluation of the manuscript revision.